# Learning Holistic-Componential Prompt Groups for Micro-Expression Recognition

## Abstract

Micro-expressions (MEs) are facial muscle movements that reveal genuine underlying emotions. Due to their subtlety and visual similarity, micro-expression recognition (MER) presents significant challenges. Existing methods mainly rely on low-level visual features and lack an understanding of high-level semantics, making it difficult to differentiate fine-grained emotional categories effectively. Facial action units (AUs) provide local action region encodings, which help establish associations between emotional semantics and action semantics. However, the complex cross-mapping relationship between emotional categories and AUs easily leads to semantic confusion. To address these problems, we propose a novel framework for MER, called HCP_MER, which leverages the powerful alignment capabilities of visual-language models such as CLIP to construct multimodal visual-language alignments through holistic-componential prompt groups. We provide corresponding holistic emotion and componential AU prompts for each emotion category to eliminate semantic ambiguity. By aligning optical flow and motion magnification representations with componential and holistic prompts, respectively, our approach establishes multi-granularity complementary visual-semantic associations. To ensure the precise attribution of predicted emotional semantics, we design a consistency constraint to enhance decision stability. Finally, we integrate adaptive gated fusion of complementary responses with downstream supervisory signal optimization to achieve fine-grained emotion discrimination. Experimental results on CASME II, SAMM, SMIC, and CAS(ME)[3] demonstrate that HCP_MER achieves competitive performance, exhibiting remarkable robustness and discriminability.

## 1 Introduction

Micro-expressions (MEs), which are brief and subtle facial movements produced when humans suppress their true emotions, have significant applications in fields such as clinical psychological diagnosis, security screening, and intelligent human-computer interaction (Oh et al., 2018b). Their extremely short duration, low-intensity localized muscle changes, and highly similar visual patterns together pose the core challenge in micro-expression recognition (MER) (Ekman & Friesen, 1969; Shen et al., 2012; Svetieva & Frank, 2016). Existing methods primarily rely on convolutional neural networks (CNNs) to extract visual features from facial images (Zhang et al., 2018b; Tran et al., 2021) or use graph neural networks (GNNs) to model facial structural information (Lei et al., 2020), achieving impressive performance. However, these approaches are limited to low-level visual features and lack the ability to understand higher-level emotional semantics, making it difficult for them to achieve fine-grained classification in emotional categories with highly similar visual features.

Recently, visual-language large models (such as CLIP (Radford et al., 2021)) have mapped images and text into a shared semantic space through large-scale contrastive learning, enabling the visual encoder to perceive rich cross-modal semantic information, which has opened up new research avenues for MER. However, the original CLIP uses a uniform and rigid template, "a photo of [class]," which is ill-suited to the local and subtle nature of MEs. A natural remedy is to introduce detailed Action Units (AUs) (Prince et al., 2015) prompts to provide finer-grained textual semantics, thereby enhancing local perception Liu et al. (2025b). Yet, as shown in Fig. 1(a), different MEs may trigger similar AU combinations, while the same AU patterns can correspond to different MEs. This many-to-many mapping implies that relying solely on AU-based semantics risks cross-category contamination in

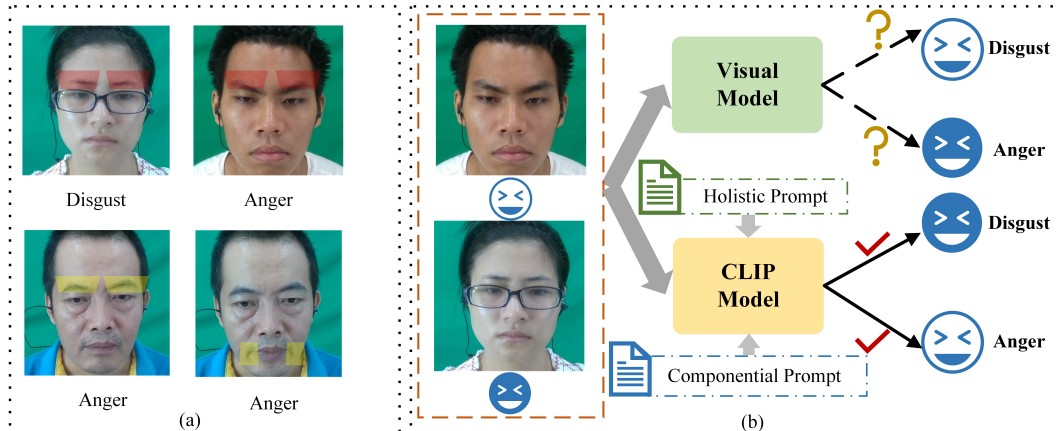

Figure 1: (a) Illustration of the cross-mapping between emotions and AUs. (Top) Many-to-one mapping: AU4 (brow lowering) serves as a shared indicator of both Anger and Disgust, revealing the inherent ambiguity of inferring emotions from local facial actions. (Bottom) One-to-many mapping: even within the same emotion category (Anger), different AU configurations may arise—e.g., AU4 (brow lowering) versus AU14 (dimpler)—highlighting the diversity of MEs. (b) Comparison between methods. (Top) Existing single visual approaches struggle to distinguish MEs with similar visual patterns, such as Anger and Disgust. (Bottom) Our method introduces emotion-bound holistic–componential prompts, providing complementary semantic context and enabling more accurate MER.

the semantic space. This key observation motivates us to jointly capture component-level semantics that reveal subtle movements and holistic semantics that convey global emotions (Fig. 1(b)).

Inspired by this fact, we propose a multimodal visual-language alignment framework for MER, called HCP_MER, which is constructed with holistic-componential prompt groups (HCP Groups). Specifically, we construct a one-to-one corresponding holistic-component prompt group for each emotional category, where the holistic prompt describes the macro emotional state, and the component prompt refines the corresponding AU combinations. This binding design addresses the complex mapping relationship between emotion and AUs, thus eliminating semantic ambiguity in single prompts. Additionally, we design a multimodal visual-language alignment: aligning the enlarged full-face image features with the holistic prompt and aligning optical flow features with the component prompt. By establishing multi-granularity complementary visual-semantic associations, we further enhance the model's sensitivity to fine-grained emotional discrimination. Furthermore, we introduce a lightweight Adapter after the visual encoder to improve cross-modal alignment quality and effectively mitigate the overfitting risk caused by the scarcity of ME data. To ensure the accurate attribution of predicted emotional semantics, we design a consistency constraint to enhance decision stability. Finally, by combining adaptive gated fusion of complementary responses and downstream supervision signal optimization, HCP_MER achieves fine-grained and robust emotional discrimination.

## 2  RELATED WORK

**MER Methods.** In early studies, researchers primarily relied on handcrafted feature extractors to capture facial expression variations across spatial and temporal dimensions. Methods such as Pfister et al. (2011) and Wang et al. (2014) were widely used to model video sequences but incurred significant computational overhead. To address this, Davison et al. (2018) proposed a novel approach that performs recognition based only on the apex and onset frames. By combining local optical flow magnitudes with global optical strain through a dual-weighting mechanism, their method effectively enhanced feature representation. However, traditional feature engineering methods, due to their inherently linear nature, struggle to capture the nonlinear and localized motion patterns characteristic of MEs. This limitation has driven the community towards deep learning frameworks,

which can learn more discriminative representations. For instance, Gan et al. (2019) and Van Quang et al. (2019) utilized apex frames and optical flow to extract structurally aware features through convolutional or capsule-based architectures, improving responsiveness to subtle facial movements. Subsequently, recurrent convolutional networks (Xia et al., 2020) introduced temporal dependencies across frames to better capture ME evolution. Moreover, the incorporation of Transformer modules has further improved the modeling of subtle movements in key facial muscle regions (Wang et al., 2024). More recently, methods such as Micro-BERT (Nguyen et al., 2023) and SelfME (Fan et al., 2023) adopted self-supervised paradigms, enabling models to inherently learn to capture the fine-grained dynamics of MEs, thus further improving classification performance.

**Vision-Language Model.** In parallel, vision-language models (VLMs) have garnered increasing attention due to their powerful multimodal semantic alignment and transfer capabilities. CLIP (Radford et al., 2021), a prominent model in this domain, maps images and text into a shared semantic space through large-scale contrastive learning on image-text pairs, achieving impressive zero-shot generalization. To enhance CLIP's adaptability to specific tasks, Zhou et al. (2022b) proposed learnable contextual prompt vectors, enabling efficient few-shot transfer without fine-tuning the backbone network. Zhou et al. (2022a) further introduced a conditional context optimization mechanism, allowing prompts to dynamically adjust based on image features to mitigate class distribution shifts. Subsequent works, such as (Gao et al., 2024) and (Tian et al., 2024), employed feature adapters and attribute-guided mechanisms to improve CLIP's performance in downstream tasks. Khattak et al. (2023) advanced this line of research by designing shared and modality-specific prompt structures and incorporating multi-layer cooperative alignment across visual and textual branches, significantly improving cross-modal consistency. In the context of facial expression recognition (FER), Ma et al. (2025) introduced a hierarchical prompt generator and a soft-hard prompt alignment strategy, which effectively alleviated semantic mismatches across modalities and led to notable improvements in cross-dataset emotion recognition. Although VLMs have demonstrated promising results in general vision tasks and FER, their application to MER is still in its early stages. As a pioneering work, Liu et al. (2025b) encoded facial AUs into semantic prompts and aligned them with CLIP's visual representations, enabling the model to learn more discriminative ME features. This work demonstrated the potential of VLMs in MER tasks. However, a key challenge remains in effectively utilizing language prompts to model the complex mappings between AUs and emotional categories.

## 3 PROPOSED METHODOLOGY

### 3.1 OVERVIEW

We propose a novel MER method, HCP_MER, whose overall framework is shown in Fig. 2. First, we construct HCP Groups, effectively addressing the emotional semantic ambiguity caused by a single AU by establishing a binding holistic-componential semantic context. Next, we design a multimodal visual-language alignment mechanism, enhancing the model's sensitivity to fine-grained emotional differences by establishing multi-granularity complementary visual-semantic associations. Furthermore, inspired by the concept of mutual distillation, we introduce a consistency constraint between the holistic and component responses to ensure stable emotional category attribution. Finally, we combine adaptive gated networks to fuse complementary responses and optimize downstream supervision signals, enabling HCP_MER to achieve fine-grained and robust emotional discrimination.

### 3.2 HCP GROUPS

Previous coarse-grained textual prompts, such as "a photo of [class]", are inadequate for precisely differentiating ME categories. Although incorporating Action Units (AUs) can refine prompts to a finer granularity, the cross-mapping between emotion categories and their associated AU codes makes emotion semantics difficult to disentangle. Consequently, we construct holistic-componential prompt groups (HCP_Groups) to address the semantic overlap induced by single-prompt designs. Concretely, we build category-specific prompt formulations and adopt the idea of COOP(Zhou et al., 2022b), enabling learnable text templates that help CLIP better adapt to downstream tasks. We define two related yet distinct templates for holistic and componential prompts. Using CLIP's tokenizer, we obtain learnable holistic context tokens $[l_1^h, l_2^h, \ldots, l_k^h]$ and learnable component context tokens $[l_1^c, l_2^c, \ldots, l_k^c]$, and introduce CLASSM as a unified class token indicating MEs. However,

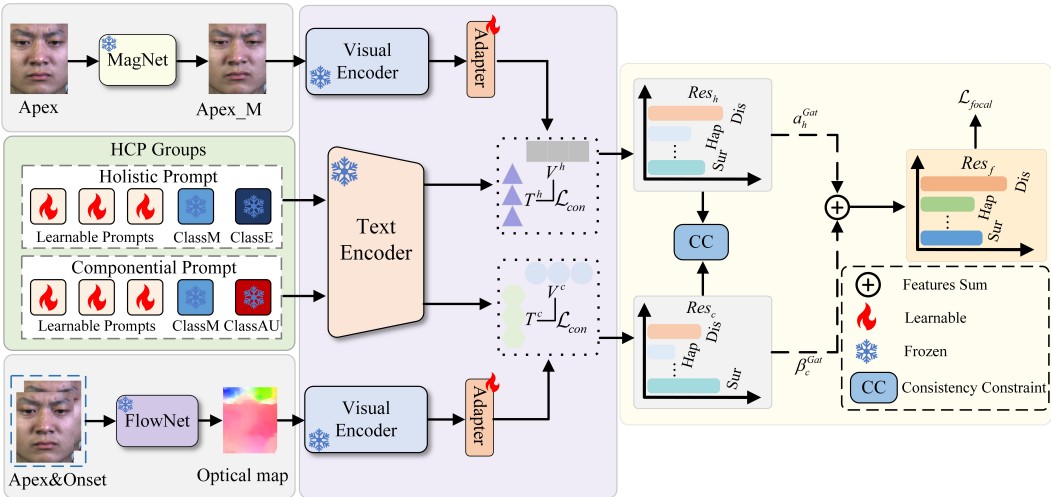

Figure 2: The overall architecture of HCP_MER. Gray blocks indicate visual inputs (Apex_M and the optical map), green blocks denote textual inputs (HCP Groups), purple blocks represent multimodal vision–language alignment, and yellow blocks perform consistency constraints with adaptive gated fusion. Dashed boxes explain the key symbols.

a single CLASSM-based prompt is still insufficient for fine-grained MER. To alleviate this issue, inspired by the attribute-guided prompt adjustment strategy in ArGue(Tian et al., 2024), we incorporate ME category-specific visual attributes to further refine the text prompts.

For the holistic prompt, we add CLASSE as an indicator of the emotion superclass so that the model can perceive the macro-level affective state of facial expressions. For the component prompt, we introduce CLASSAU, which denotes the specific AU combination corresponding to each ME category and encodes the localized motion regions. With the tokenizer, we convert $[class]$ into a class token as follows:

$$T_c = \text{tokenizer}([CLASS]), \tag{1}$$

We then expand $T_c$ into three class tokens for CLASSM, CLASSE, and CLASSAU, denoted as $T_c^m$, $T_c^e$, and $T_c^{au}$, respectively. This yields the complete holistic prompt sequence $P_h$ and component prompt sequence $P_c$:

$$\begin{aligned}
P_h &= \{l_1^h, \ldots, l_{\lfloor k/2 \rfloor}^h, T_c^m, T_c^e, l_{\lfloor k/2 \rfloor+1}^h, \ldots, l_k^h\}, \\
P_c &= \{l_1^c, \ldots, l_{\lfloor k/2 \rfloor}^c, T_c^m, T_c^{au}, l_{\lfloor k/2 \rfloor+1}^c, \ldots, l_k^c\}.
\end{aligned} \tag{2}$$

We insert the class tokens $\{T_c^m, T_c^e, T_c^{au}\}$ into the middle of $P_h$ and $P_c$, and feed them into the pretrained CLIP text encoder to obtain high-dimensional semantic embeddings:

$$T^h = \tau(P_h) \qquad T^c = \tau(P_c), \tag{3}$$

where $\tau$ denotes the text encoder, and $T^h \in \mathbb{R}^{B \times N}$ and $T^c \in \mathbb{R}^{B \times N}$ are the holistic and component semantic embeddings, respectively. Here, $B$ is the batch size and $N$ is the embedding length.

With this design, each holistic prompt is explicitly paired with a corresponding component prompt, forming an HCP Groups. The holistic prompt provides semantic context for the component AUs to disambiguate visually similar AU patterns, while the component prompt supplies fine-grained cues for the holistic emotion to capture diverse manifestations within the same affective class. As a result, semantic overlap across different emotion categories is effectively reduced (Appendix B provides an example of the HCP Groups prompts for a specific emotion category.)

### 3.3 MULTIMODAL VISUAL-LANGUAGE ALIGNMENT

We know that the original CLIP uses the entire image as the visual feature input. Although this performs excellently on natural datasets such as ImageNet, MEs exhibit highly similar facial back-

grounds and extremely low motion intensity, causing the visual features to show high similarity, which reduces the model's discriminative sensitivity. Furthermore, directly fine-tuning the entire visual encoder would inevitably disrupt the original pre-trained knowledge while also facing severe overfitting risks due to the scarcity of ME data.

To address this, we propose Multimodal Visual-Language Alignment. In terms of visual input, we use the classic MagNet magnification algorithm (Oh et al., 2018a) to obtain the motion-magnified apex frame (Apex_M), which helps highlight the muscle changes occurring during MEs. Simultaneously, the optical flow estimation algorithm, FlowNet (Ilg et al., 2017), computes the motion information between the starting frame and the apex frame, generating an optical flow map to further enhance the capture of temporal motion information. This approach increases the visual saliency difference of MEs in both spatial and spatiotemporal dimensions.

We align the Apex_M visual features with the holistic emotional text description, enabling the model to construct the overall emotional semantics. Meanwhile, we align the optical flow map with the componential AU text description, forcing the model to focus on the semantics of local subtle motions. Through Multimodal Visual-Language Alignment, we establish multi-granularity complementary visual-semantic associations, further enhancing the model's sensitivity to fine-grained emotions. Moreover, to ensure efficient adaptation of MEs to CLIP and reduce the risk of model overfitting, we add a lightweight adapter after the visual encoder (detailed architecture in Appendix B), and combine cosine similarity matching with contrastive loss to further optimize the cross-modal alignment quality. The specific formula is as follows:

$$S(V, T) = \sum_{k=h,c} \frac{V^k \cdot T^k}{\|V^k\|\|T^k\|}, \tag{4}$$

$$\mathcal{L}_{\text{con}} = - \sum_{i=h,c} \log \left( \frac{\exp(S(V^i, T^i)/\tau)}{\exp(S(V^i, T^i)/\tau) + \sum_{j \neq i} \exp(S(V^i, T^j)/\tau)} \right), \tag{5}$$

where $V^k$ and $T^k$ represent the holistic and componential visual features and text features, respectively. $\|V^k\|$ and $\|T^k\|$ are the norms of the visual and text features, respectively. $S(V^i, T^i)$ represents the similarity between the visual feature $V^i$ and the text feature $T^i$, while $S(V^i, T^j)$ represents the similarity between different visual and text features. $\tau$ represents the temperature parameter, which is used to adjust the sensitivity of the similarity.

## 3.4 CONSISTENCY CONSTRAINT

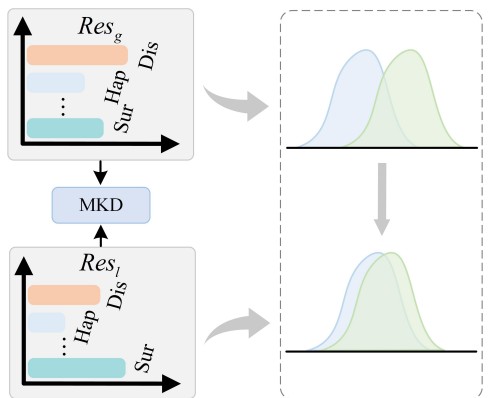

Figure 3: Illustration of the consistency constraint.

Although our approach leverages multi-granularity complementary visual-semantic representations to enhance sensitivity to fine-grained distinctions, the holistic and component branches inherently rely on distinct visual cues and textual contexts. These semantic differences may lead the model to learn inconsistent emotional features, as reflected in the distribution discrepancies between holistic response $Res_h$ and componential response $Res_c$, which in turn affects the accurate attribution of emotional categories. We aim to maintain the model's sensitivity to subtle emotional differences while improving decision stability. To address this, we introduce a consistency constraint (CC) mechanism based on mutual knowledge distillation (MKD) (Zhang et al., 2018a), as shown in Fig. 3.

The CC mechanism bridges $Res_h$ and $Res_c$, facilitating knowledge sharing between them, ensuring consistency in their response space distributions, and avoiding semantic confusion. The specific

formula is as follows:

$$\mathcal{L}_{\text{JS}} = \frac{1}{2}\left[\sum_i Res_h(i) \log\left(\frac{Res_h(i)}{Res_c(i)}\right) + \sum_i Res_c(i) \log\left(\frac{Res_c(i)}{Res_h(i)}\right)\right], \qquad (6)$$

Here, $Res_h(i)$ and $Res_c(i)$ represent the holistic and componential responses for the $i$-th category, respectively. We treat the output probability distributions of the overall and component branches as soft labels for each other and use symmetric KL divergence as the distribution consistency measure, thereby achieving more robust emotional semantic prediction.

## 3.5 ADAPTIVE GATED FUSION

We introduce a gated network to adaptively fuse multi-granularity complementary response outputs, selecting the optimal information output with minimal additional parameter cost. We define the gating function to calculate the weights for the holistic and component responses, specifically as follows:

$$w^{Gat} = \sigma(W[Res_h; Res_c]) + b, \qquad (7)$$

$$Res_f = a_h^{\text{Gat}} \cdot Res_h + \beta_c^{\text{Gat}} \cdot Res_c, \qquad (8)$$

where $\sigma(\cdot)$ denotes the sigmoid activation function, and $W$ and $b$ represent the learned weight and bias. This yields the weights $a_h^{\text{Gat}}$ and $\beta_c^{\text{Gat}}$ for the holistic and component responses, respectively. After the weighted combination, the final response distribution $Res_f$ is obtained and combined with the downstream class imbalance loss, focal loss, for supervised optimization. The weights are updated to the optimal ratio. Ultimately, HCP_MER achieves fine-grained and robust emotional discrimination, and the total loss function is composed as follows:

$$\mathcal{L}_{\text{final}} = \mathcal{L}_{\text{con}} + \lambda_1 \mathcal{L}_{\text{JS}} + \lambda_2 \mathcal{L}_{\text{focal}}, \qquad (9)$$

where $\mathcal{L}_{\text{con}}$ represents the contrastive loss, $\mathcal{L}_{\text{JS}}$ represents the KL divergence loss, and $\mathcal{L}_{\text{focal}}$ represents the focal loss. $\lambda_1$ and $\lambda_2$ are the corresponding hyperparameters.

## 4 EXPERIMENTS

### 4.1 EXPERIMENTAL CONFIGURATION

**Implementation Details.** The detailed experimental implementation settings are provided in Appendix C. Moreover, to further ensure fair comparisons across different models and to avoid evaluation bias caused by subject-specific individual differences, we adopt the Leave-One-Subject-Out (LOSO) cross-validation protocol for model training and assessment.

**Experimental Metrics.** Considering the class imbalance in the ME datasets, accuracy as a traditional evaluation metric may not fully reflect the model's performance. Therefore, in addition to accuracy, we also introduce the unweighted F1 score (UF1) and the unweighted average recall (UAR) as supplementary experimental metrics. These are used to evaluate the effectiveness of the model, and a detailed explanation of these metrics can be found in Appendix C.

### 4.2 COMPARISON WITH STATE-OF-THE-ART METHODS

We conducted comparative experiments with state-of-the-art methods on the SMIC Li et al. (2013), CASME II(Yan et al., 2014), SAMM (Davison et al., 2016), and CAS(ME)³ (Li et al., 2022) datasets, on which we performed single-dataset evaluations, while cross-database evaluations were conducted between CASME II(Yan et al., 2014) and SAMM (Davison et al., 2016). The detailed configuration of the datasets can be found in Appendix C.

**Results on SMIC, CASME II, and SAMM.** As shown in Tab. 1, our comparison methods include both traditional handcrafted feature-based approaches and deep learning methods. HCP_MER achieves competitive or the best performance across all three datasets. Notably, on CASME II

Table 1: Comparative experimental results for 3-class task (UF1 and UAR on the SMIC, CASME II, and SAMM Datasets).

| Methods | SMIC | | CASME II | | SAMM | |
|---|---|---|---|---|---|---|
| | UF1 | UAR | UF1 | UAR | UF1 | UAR |
| LBP-TOP(Pfister et al., 2011) | 0.2000 | 0.5280 | 0.7026 | 0.7429 | 0.3957 | 0.4102 |
| Bi-WOOF(Davison et al., 2018) | 0.5727 | 0.5829 | 0.7805 | 0.8026 | 0.5211 | 0.5139 |
| CapsuleNet(Van Quang et al., 2019) | 0.5820 | 0.5877 | 0.7068 | 0.7018 | 0.6209 | 0.5989 |
| OFF-ApexNet(Gan et al., 2019) | 0.6817 | 0.6950 | 0.8764 | 0.8681 | 0.5409 | 0.5392 |
| RCN(Xia et al., 2020) | 0.6326 | 0.6441 | 0.8512 | 0.8123 | 0.7601 | 0.6715 |
| ICE-GAN(Yu et al., 2021) | 0.5727 | 0.5829 | 0.7805 | 0.8026 | 0.5211 | 0.5139 |
| SLSTT(Zhang et al., 2022) | 0.7240 | 0.7070 | 0.9010 | 0.8850 | 0.7150 | 0.6420 |
| FeatRef(Zhou et al., 2022c) | 0.7011 | 0.7083 | 0.8911 | 0.8873 | 0.7372 | 0.7155 |
| ME-PLAN(Zhao et al., 2022) | 0.7130 | 0.7260 | 0.8630 | 0.8780 | 0.7160 | 0.7420 |
| Micro-BERT(Nguyen et al., 2023) | - | - | 0.9034 | 0.8914 | - | - |
| SelfME(Fan et al., 2023) | - | - | 0.9078 | 0.9230 | - | - |
| HTNet(Wang et al., 2024) | **0.8049** | 0.7905 | 0.9532 | 0.9516 | 0.8131 | 0.8124 |
| EMRNet(Liu et al., 2025a) | 0.6509 | 0.6596 | 0.9074 | 0.8995 | 0.6782 | 0.6897 |
| MER-CLIP(Liu et al., 2025b) | - | - | 0.9409 | 0.9487 | 0.8321 | 0.8434 |
| **Ours** | 0.8032 | **0.8146** | **0.9547** | **0.9560** | **0.8426** | **0.8593** |

Table 2: Comparative experimental results for 3-class task (UF1 and UAR on the CAS(ME)$^3$ Dataset).

| Methods | CAS(ME)$^3$ | |
|---|---|---|
| | UF1 | UAR |
| AlexNet(Zhang & Zhang, 2022) | 0.2570 | 0.2634 |
| RCN-A(Xia et al., 2020) | 0.3928 | 0.3893 |
| FearRef(Zhou et al., 2022c) | 0.4930 | 0.3413 |
| u-bert(Nguyen et al., 2023) | 0.5604 | 0.6125 |
| HTNet(Wang et al., 2024) | 0.5767 | 0.5415 |
| FDP(Shao et al., 2025) | 0.5978 | 0.5784 |
| MER-CLIP(Liu et al., 2025b) | 0.7832 | 0.7606 |
| **Ours** | **0.8052** | **0.8012** |

and SAMM, our model outperforms all previous methods with UF1/UAR of **0.9547/0.9560** and **0.8426/0.8593**, respectively. On SMIC, our method achieves the highest UAR, demonstrating excellent discriminability. However, HTNet achieves the highest UF1 on SMIC, reflecting the advantages of the transformer architecture in modeling ME features. Our method, by balancing the retention of pre-trained knowledge and mitigating overfitting risks, adopts a frozen visual encoder with an adapter, which slightly limits the performance ceiling.

**Results on CAS(ME)$^3$.** As shown in Tab. 2 and Tab. 3, we conducted extended experiments on the more challenging CAS(ME)$^3$ dataset. In the 3-class classification task, we achieved UF1/UAR of **0.8052/0.8012**. In the 4-class classification task, we achieved **0.7168/0.6996**. In the finest-grained 7-class classification task, we reached **0.5955/0.6047**, outperforming MER-CLIP by **+9.58%** and **+10.33%**, respectively. As the number of classes increases, the performance of traditional or single-modal methods rapidly declines. In contrast, our method endows the model with the ability to understand high-level emotional semantics and component semantics. The combination of HCP Groups effectively resolves the confusion caused by overlapping emotional and action semantics.

Table 3: Comparative experimental results for 4-class and 7-class tasks (UF1 and UAR on the CAS(ME)$^3$ Dataset).

| Methods | CAS(ME)$^3$ | | | |
| | 4-CLASS | | 7-CLASS | |
| | UF1 | UAR | UF1 | UAR |
|---|---|---|---|---|
| AlexNet(Zhang & Zhang, 2022) | 0.2915 | 0.2910 | 0.1759 | 0.1801 |
| SFAMNet(Liong et al., 2024) | 0.4462 | 0.4797 | 0.2365 | 0.2373 |
| u-bert(Nguyen et al., 2023) | 0.4718 | 0.4913 | 0.3264 | 0.3254 |
| ATM-GCN(Zhang et al., 2024) | 0.5423 | 0.5330 | 0.4308 | 0.4283 |
| MER-CLIP(Liu et al., 2025b) | 0.6544 | 0.6242 | 0.4997 | 0.5014 |
| **Ours** | **0.7168** | **0.6996** | **0.5955** | **0.6047** |

Table 4: Comparative experimental results for Cross-database evaluation (ACC and UAR on the CASME II Dataset).

| Methods | CASMEII→SAMM | | SAMM→CASMEII | |
| | ACC | UAR | ACC | UAR |
|---|---|---|---|---|
| LBP-TOP(Pfister et al., 2011) | 0.3380 | 0.3270 | 0.2320 | 0.3160 |
| 3DHOG(Polikovsky et al., 2009) | 0.3530 | 0.2690 | 0.3730 | 0.1870 |
| MDMO(Liu et al., 2015) | 0.4410 | 0.3490 | 0.2650 | 0.3460 |
| I$^2$-Transformer(Shao et al., 2023) | 0.5120 | - | **0.6620** | - |
| FDP(Shao et al., 2025) | **0.5820** | 0.5180 | 0.6220 | **0.5600** |
| **Ours** | 0.5680 | **0.5570** | 0.6430 | 0.5330 |

Additionally, the multimodal visual-language alignment builds a complementary, multi-granular visual semantic relationship, while CC further enhances decision stability and adaptive gated fusion for fine-tuned response outputs. As a result, HCP_MER maintains stable and superior performance in more complex emotional spaces.

**Generalization Evaluation Results.** To assess the generalization capability of our model across different micro-expression datasets, we conduct cross-dataset evaluations on CASME II and SAMM with two transfer directions: training on CASME II and testing on SAMM (CASME II→SAMM), and training on SAMM and testing on CASME II (SAMM→CASME II). We report Accuracy (ACC) and Unweighted Average Recall (UAR) as evaluation metrics. As shown in Table X, our proposed HCP_MER achieves competitive performance under both transfer settings and attains the highest UAR of 0.5570 in the CASME II→SAMM setting. These results indicate that, compared with purely visual approaches, our method benefits from the high-level emotional semantics introduced by the HCP_Groups and further enhances emotion discriminability through multi-granularity complementary semantic alignment, effectively mitigating domain shift caused by differences in frame rate, subject ethnicity, and annotation protocols. Although the frozen visual encoder and the limited data scale impose certain constraints on performance improvement, the cross-dataset results demonstrate the robustness and strong generalization ability of our approach in challenging domain-transfer scenarios.

## 4.3 ABLATION STUDY

We systematically evaluate the contribution of each module on the CAS(ME)$^3$ dataset across 3-class, 4-class, and 7-class tasks. The full model HCP_MER demonstrates excellent performance in all tasks: 3-class (UF1/UAR = **0.8052/0.8012**), 4-class (UF1/UAR = **0.7168/0.6996**), and 7-class (UF1/UAR = **0.5955/0.6247**). The ablation experiments, as shown in the Fig. 4, reveal that using only the holistic branch (w/o COM) results in a decrease of UF1 to **0.6515** in the 3-class task, indicating that the lack of local AU details severely weakens the model's ability to capture subtle move-

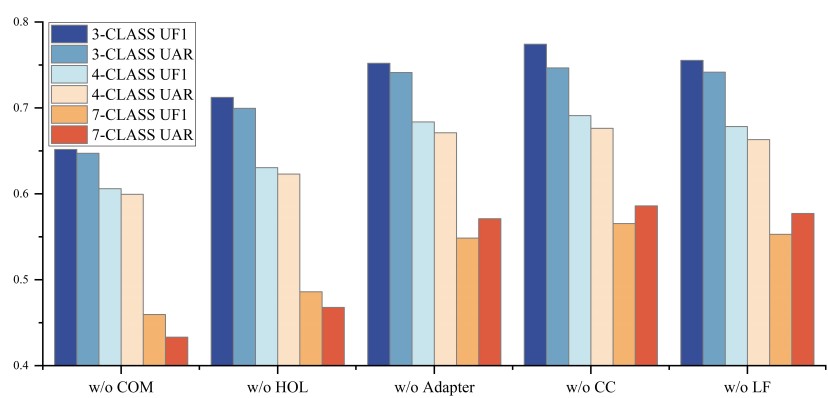

Figure 4: Ablation study on the contributions of different components in HCP_MER.

ments. When only the componental branch is used (w/o HOL), UF1 drops to **0.7123**, suggesting that the absence of global emotional context leads to incomplete semantics and classification difficulties. Removing the Adapter module (w/o Adapter) causes a significant performance drop across all tasks, highlighting its key role in retaining the pre-trained knowledge from CLIP and enhancing the alignment quality between the ME domain and textual semantics. Removing the Consistency Map (w/o CC) and $\mathcal{L}_{\text{focal}}$ (w/o LF) leads to decreased prediction stability and exacerbates the class imbalance problem, especially in the 7-class task. The experiments fully demonstrate that the HCP Groups and multi-visual-language alignment are the core components of HCP_MER. These modules work effectively together to enhance the model's discriminative power and robustness.

## 4.4 VISUALIZATION

**Feature Distribution Visualization.** We further employed t-SNE to analyze the feature distributions across different configurations on the 7-class task of the CAS(ME)[3] dataset. In the baseline model without the Adapter (a), the feature distribution is highly mixed, highlighting that the pre-trained CLIP weights alone are insufficient for the MER task. The adapter bridges the gap between visual and textual features, improving alignment. Adding the Adapter without CC and $\mathcal{L}_{\text{focal}}$ (LF) for decision consistency and class imbalance handling (b) improves inter-class separability, although significant overlap persists. In contrast, our proposed HCP_MER method (c) substantially enhances the feature space's geometric structure: samples from the same class form compact clusters, while those from different classes are clearly separated. The method also improves discriminability, particularly for semantically similar categories. This confirms the effectiveness of our approach in fine-grained MER, aligning with our quantitative results.

**Visualization of Attention Distributions.** We present a visual analysis of the attention distributions across the holistic and componental branches for various emotional samples. As shown in Fig. 6, the two branches exhibit distinctly different yet complementary attention patterns. Specifically, the holistic branch demonstrates a broad, diffuse attention distribution, typically spanning macro facial regions crucial for understanding the overall emotional context. For example, for the happy emotion, we observed that the attention covers the cheeks, eyes, and lips, which are key areas associated with the macroscopic expression of happiness. At the same time, the component branch shows highly localized and concentrated attention, focusing on specific muscle groups related to AU activations. For instance, the attention corresponding to the happy emotion is predominantly concentrated around the eyes and lips, which reflects the fine-grained componental semantics.

We observe that the attention distributions of the two branches exhibit complementary characteristics. This clear divergence in attention patterns verifies that our HCP Groups successfully guide the visual encoder to perceive semantically distinct yet complementary features. Simultaneously, the presence of overlapping attention areas indicates that the model performs collaborative observation of the same facial regions from different semantic dimensions. Based on these characteristics, the adaptive gated fusion network does not simply merge these features, but rather learns to dynamically recalibrate and assign optimal weights according to the input sample. This process is further

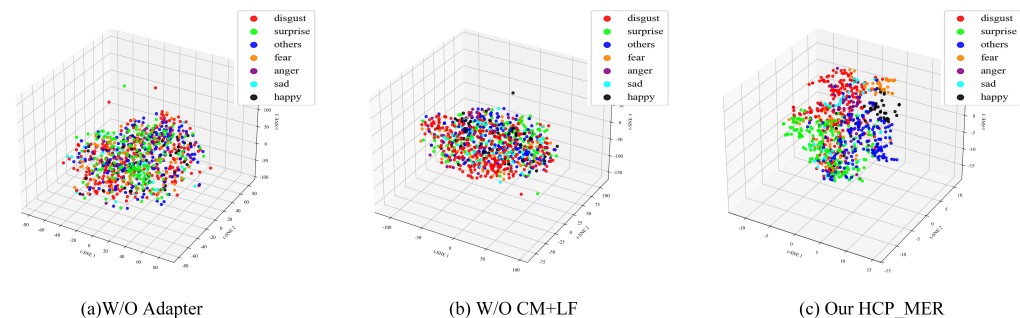

(a)W/O Adapter        (b) W/O CM+LF        (c) Our HCP_MER

Figure 5: Feature space distribution for the 7-class classification task on CAS(ME)$^3$. Features are extracted from the fused representation $Res_f$ and projected to 3D space using t-SNE.

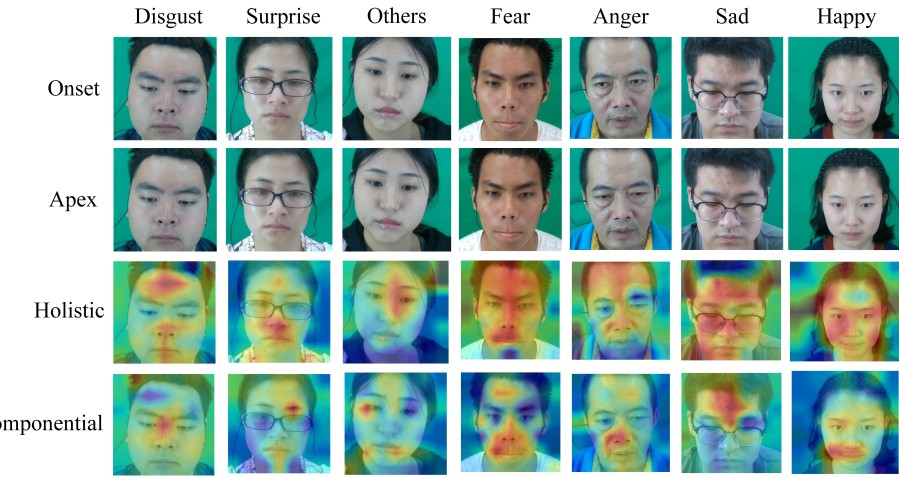

Figure 6: Attention distribution across the holistic and componential branches. Spatial attention maps are derived from the [CLS]-to-patch attention weights in the last Transformer layer of CLIP's Visual encoder, upsampled to the original resolution.

optimized under the guidance of downstream supervisory signals, enabling the model to execute refined weight allocation between macroscopic expression context and subtle motion details, thereby achieving more accurate emotion discrimination.

## 5 CONCLUSION

In this paper, we propose a novel MER framework, HCP_MER. We introduce the holistic-componential prompt groups, which effectively alleviate the semantic ambiguity issue by binding holistic emotional semantics with componential AUs semantics. At the same time, leveraging the powerful alignment capabilities of VLMs like CLIP, we propose a multimodal visual-language alignment approach that establishes multi-granularity complementary visual-semantic associations, enhancing the model's sensitivity to fine-grained emotional discrimination. Building on this, the consistency constraint ensures the accurate attribution of emotional predictions, while adaptive gated fusion combines complementary responses from different branches and incorporates fine-tuned optimization with downstream supervisory signals. Extensive experiments validate the superiority of our method, demonstrating the robustness and discriminative power advantages of HCP_MER.

Our method provides a new research perspective for fine-grained MER based on VLMs. In the future, we will leverage the powerful generative capabilities of LLMs or MLLMs to further explore mechanisms for the automatic generation of textual prompts.

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

APPENDIX

The appendix is structured as follows:

- Appendix A elucidates and visualizes the cross-mapping problem between emotion categories and AU units.
- Appendix B presents the implementation details of the proposed HCP Groups and Adapter.
- Appendix C provides additional information on the experimental setup and results.
- Appendix D provides details on the use of LLMs.

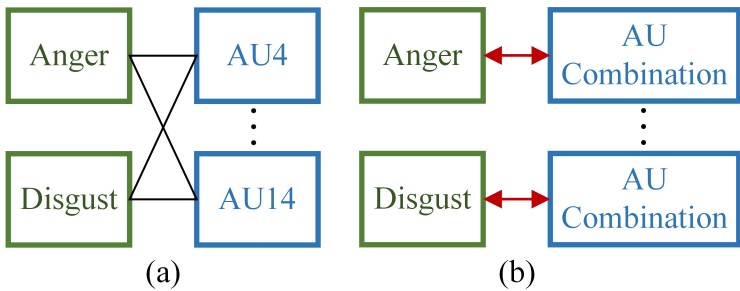

Figure 7: (a) Single AU prompting method, (b) Our proposed HCP Groups.

## A  CROSS MAPPING

For the cross-mapping relationships mentioned in the introduction, as illustrated in Fig. 7(a), we aim not to simply provide an independent AU prompt for each emotion category, but instead to construct a one-to-one set of holistic-componential prompt groups for each emotion category, as shown in Fig. 7(b). Specifically, for example, for the *anger* category in the 7-class setting of CAS(ME)[3], the holistic prompt takes the form "a photo of [CLASSM] [CLASSE]", where [CLASSM] is "micro-expression of" and [CLASSE] is *anger*. The component prompt takes the form "a photo of [CLASSM] [CLASSAU]", where [CLASSM] is "micro-expression of" and [CLASSAU] is "A combination of lowering and drawing the brows together, pressing the lips firmly, and sometimes flaring the nostrils." By establishing a binding between the holistic emotion and its component AUs, the holistic prompt provides semantic context for the component AUs to distinguish similar AU combinations, while the component prompt offers fine-grained information for the holistic emotion to capture diverse manifestations of the same emotion.

## B  IMPLEMENTATION DETAILS

### B.1  HCP GROUPS

To clearly illustrate the construction logic of HCP_Groups, we present it in the form of pseudocode.

---

**Algorithm 1** HCP Groups Construction

---

**Input:** Template " a photo of ", content of [CLASS].
**Output:** High-dimensional embeddings $T^h, T^c$.
Initialize learnable structured template.
Define holistic and component prompt sequences: $P_h = [l_1^h, \ldots, l_k^h]$, $P_c = [l_1^c, \ldots, l_k^c]$.
Define class token $t_c = \text{tokenizer}[CLASS]$.
Expand class token into: $t_c^m, t_c^e, t_c^{au}$.
Get three token classes: $t_c^m, t_c^e, t_c^{au}$.
**for** each emotion category and AU combination **do**
    Insert $t_c^m, t_c^e, t_c^{au}$ into $P_h$ and $P_c$.
    Update $P_h$ and $P_c$ with token classes:
    $P_h = [l_1^h, \ldots, t_c^m, t_c^e, \ldots, l_k^h]$
    $P_c = [l_1^c, \ldots, t_c^m, t_c^{au}, \ldots, l_k^c]$
    Apply CLIP tokenizer: $T^h = \tau(P_h), \quad T^c = \tau(P_c)$
**end for**
**Return** $T^h, T^c$

---

### B.2  ADAPTER DESIGN

To mitigate the risk of overfitting in MER, we incorporate a lightweight adapter module after the visual encoder. As illustrated in Fig. 8, the adapter follows a residual design. Specifically, the extracted features are first projected into a lower-dimensional space through a linear layer, followed

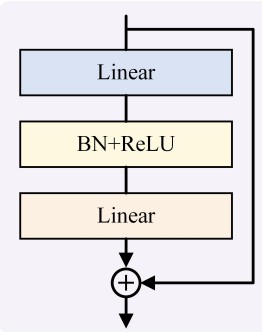

Figure 8: The design of the adapter.

Table 5: Number of Samples per Class for the 3-Class Task on SMIC, CASME II, and SAMM

| SMIC | | CASME II | | SAMM | |
|---|---|---|---|---|---|
| **Class** | **Num** | **Class** | **Num** | **Class** | **Num** |
| Positive | 51 | Positive | 32 | Positive | 26 |
| Negative | 70 | Negative | 90 | Negative | 92 |
| Surprise | 43 | Surprise | 25 | Surprise | 15 |

by BatchNorm and a ReLU activation for normalization and nonlinear transformation. The transformed features are then restored to the original dimensionality via another linear layer, after which the input features are added back through a residual connection.

Furthermore, we adaptively adjust the adapter's complexity according to the dataset size. For smaller datasets such as CASME II (Yan et al., 2014), SAMM (Davison et al., 2016), and SMIC (Li et al., 2013), where the number of samples is limited, we employ a single adapter layer to constrain the parameter count. In contrast, for larger datasets such as CAS(ME)$^3$ (Li et al., 2022), we adopt a multi-layer adapter structure, which increases model capacity and enhances the quality of cross-modal alignment.

## C  EXPERIMENTAL SETUP AND RESULTS

### C.1  DATASETS

In this paper, we conduct experiments using four publicly available ME datasets. The experiments are carried out for a 3-class classification task on the CASME II Yan et al. (2014), SMIC Li et al. (2013), and SAMM Davison et al. (2016) datasets, while for the CAS(ME)$^3$ Li et al. (2022) dataset, we perform 3-class, 4-class, and 7-class classification experiments. Tab. 5 and Tab. 6 present the sample sizes for each class in the different datasets, and Tab. 7 reports the number of samples per class for the cross-database evaluations on CASME II and SAMM.

The CASME II dataset consists of data from 26 subjects, with a total of 255 samples. All samples were captured in a laboratory setting with a camera at 200 fps and a resolution of $640 \times 480$ pixels. The samples span seven emotion categories: happiness, surprise, disgust, sadness, fear, repression, and others.

The SMIC dataset includes three subsets captured by different types of cameras: HS (high-speed camera), VIS (visual spectrum camera), and NIR (near-infrared camera). As high-speed cameras can effectively capture the subtle and transient changes of MEs, we selected the HS subset for our experiments. This subset contains data from 16 subjects, recorded at 100 fps with a resolution of $640 \times 480$ pixels, and includes three emotions: positive, negative, and surprise.

The SAMM dataset includes data from 28 subjects, with a total of 159 samples. All samples were recorded using high-speed cameras with a frame rate of 200 fps and a resolution of $2040 \times 1088$

Table 6: Number of Samples per Class in the 3-class, 4-class, and 7-class Tasks on CAS(ME)[3]

| | CAS(ME)[3] | |
|---|---|---|
| | **Class** | **Num** |
| **3-class** | Positive | 57 |
| | Negative | 457 |
| | Surprise | 187 |
| **4-class** | Negative | 457 |
| | Positive | 57 |
| | Surprise | 187 |
| | Others | 161 |
| **7-class** | Disgust | 250 |
| | Fear | 86 |
| | Anger | 64 |
| | Sad | 57 |
| | Happy | 57 |
| | Surprise | 187 |
| | Others | 161 |

Table 7: Number of samples per class for the cross-database evaluations on CASME II and SAMM.

| CASME II | | SAMM | |
|---|---|---|---|
| **Class** | **Num** | **Class** | **Num** |
| Happiness | 32 | Happiness | 26 |
| Others | 99 | Others | 26 |
| Surprise | 25 | Surprise | 15 |

pixels. This dataset contains eight emotions, including happiness, contempt, disgust, surprise, fear, anger, sadness, and others.

The CAS(ME)[3] dataset contains spontaneous ME videos from 216 subjects, divided into three parts: Part A includes 1,300 videos (943 MEs and 3,143 macro-expressions); Part B consists of 1,508 unlabeled videos; and Part C contains simulated crime scenario videos with high ecological validity (166 MEs and 347 macro-expressions). The dataset covers seven emotion categories: happiness, disgust, fear, anger, sadness, surprise, and others.

## C.2 EVALUATION METRICS

In this paper, we adopt three standard metrics for MER: Accuracy, Unweighted F1-score (UF1), and Unweighted Average Recall (UAR). Their formulations are given below:

$$\text{Accuracy} = \frac{\sum_{i=1}^{C} TP_i}{N} \tag{10}$$

$$UF1 = \frac{1}{C} \sum_{i=1}^{C} \frac{2 \times TP_i}{2 \times TP_i + FP_i + FN_i} \tag{11}$$

$$UAR = \frac{1}{C} \sum_{i=1}^{C} \frac{TP_i}{TP_i + FN_i} \tag{12}$$

where $C$ denotes the total number of classes, $N$ denotes the total number of samples, $TP_i$ represents the number of samples in the $i$-th class that are correctly predicted, $FP_i$ represents the number of

samples that are incorrectly predicted as the $i$-th class, and $FN_i$ represents the number of samples in the $i$-th class that are incorrectly predicted as other classes.

### C.3 EXPERIMENTAL CONFIGURATION DETAILS

To ensure reproducibility, we provide full implementation details here. All experiments are implemented in PyTorch and trained on an NVIDIA GeForce RTX3080Ti GPU. We train the model for 300 epochs using the AdamW optimizer with an initial learning rate of $1 \times 10^{-4}$ and a batch size of 16. For the loss hyperparameters, the weight of the consistency regularization $\lambda_1$ and the weight of the Focal Loss $\lambda_2$ are both set to 0.5; the Focal Loss parameters $\alpha$ and $\gamma$ are set to 0.25 and 2.0, respectively.

Regarding the architecture, we adopt CLIP ViT-B/32 as the visual encoder and freeze its pretrained weights, optimizing only the lightweight Adapter modules. For small-scale datasets, we use a single-layer Adapter with a bottleneck dimension of 64; for the large-scale CAS(ME)$^3$ dataset, we employ a two-layer Adapter with a bottleneck dimension of 128. In prompt engineering, the number of context tokens is set to $k = 8$, and the temperature parameter is $\tau = 0.01$. Visual preprocessing includes MagNet-based motion amplification with an amplification factor of 2, and optical flow is computed using a pretrained FlowNet2.0 model.

### C.4 ADDITIONAL RESULTS

To provide a more comprehensive evaluation of our method, we present the confusion matrices on several public ME datasets, including SMIC (Li et al., 2013), CASME II (Yan et al., 2014), SAMM (Davison et al., 2016), and CAS(ME)$^3$ (Li et al., 2022), as illustrated in Fig.9. In the 3-class tasks on SMIC, CASME II, SAMM, and CAS(ME)$^3$, our approach yields a high proportion of correct predictions along the diagonal, indicating strong discriminative capability. Notably, CASME II and SAMM exhibit particularly stable performance, though some confusion remains between the negative and surprise categories.

For the 4-class task on CAS(ME)$^3$, the model achieves higher accuracy on the negative and surprise categories, while the others category proves more challenging due to their inherent diversity. In the 7-class task on CAS(ME)$^3$, the model demonstrates relatively strong recognition of disgust and surprise, whereas fear, happy, and sadness are more frequently misclassified. This reflects the greater difficulty of distinguishing fine-grained emotions under conditions of sample imbalance and subtle inter-class variations.

Overall, these results not only confirm the effectiveness of HCP_MER across diverse datasets and task settings but also highlight its strong capability in discriminating emotions within complex contextual scenarios.

### C.5 SENSITIVITY ANALYSIS

To examine the robustness of HCP_MER with respect to hyperparameter choices, we conduct a sensitivity analysis on $\lambda_1$ and $\lambda_2$ in the 7-class task of CAS(ME)$^3$. Specifically, we vary one hyperparameter at a time while fixing all others to their default values used in our experiments. As shown in Fig 10, we report UAR and UF1 under $\{0.01, 0.05, 0.1, 0.5, 1\}$. The best performance is achieved at $\lambda_1 = 0.5$ and $\lambda_2 = 0.5$ (UF1 = 0.5955, UAR = 0.6247). When $\lambda_1$ is too small (0.01), the consistency regularization $\mathcal{L}_{JS}$ becomes insufficient, leading to less stable decisions on hard samples. When $\lambda_2$ is too small (0.01), the effect of $\mathcal{L}_{focal}$ is weakened, exacerbating class imbalance and degrading performance. Overall, even on this most challenging 7-class task, UF1 and UAR vary only mildly across a wide range of $\lambda_1$ and $\lambda_2$, indicating that our method is not sensitive to hyperparameter tuning and demonstrating the strong robustness of HCP_MER.

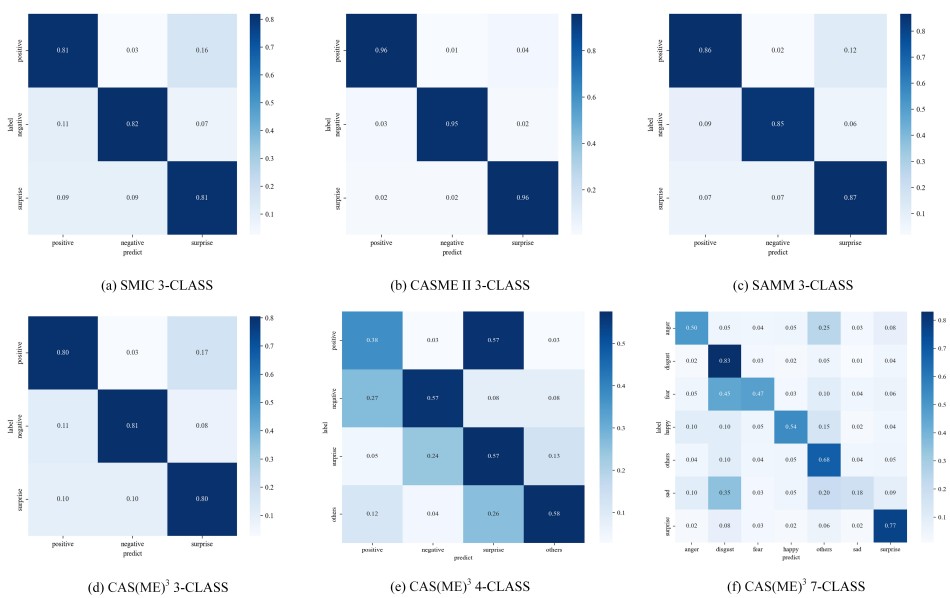

Figure 9: Confusion matrices across multiple datasets and tasks.

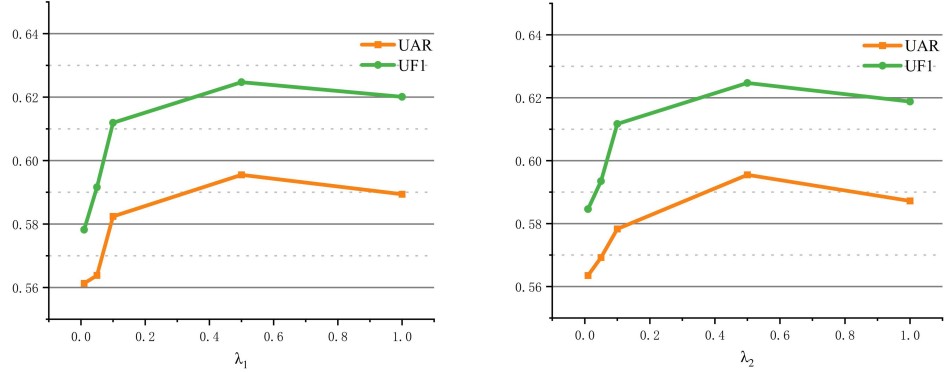

Figure 10: Sensitivity of HCP_MER to $\lambda_1$ and $\lambda_2$ on the 7-class CAS(ME)$^3$ dataset.

# D    THE USE OF LLMS

## D.1    USE OF LLMS IN RELATED WORK

We used LLMs to help search for relevant literature, in order to better evaluate prior methods and compare them with our work.

## D.2    USE OF LLMS IN WRITING

We used LLMs for translation and writing refinement so that the wording of our paper would be more standardized and appropriate.

