# OpenReview forum: "Learning Holistic-Componential Prompt Groups for Micro-Expression Recognition"
_ICLR.cc/2026/Conference — Submitted to ICLR 2026_

### Official Review · Reviewer_GpHs · 2025-10-27

**Soundness:** 2
**Presentation:** 2
**Contribution:** 2
**Rating:** 2
**Confidence:** 4

**Summary:**

The paper proposes HCP_MER, a vision–language framework for micro-expression recognition (MER) that builds Holistic–Componential Prompt (HCP) Groups per emotion class: a “holistic” prompt for macro emotion semantics and a “componential” prompt encoding AU combinations.  A lightweight adapter sits after CLIP’s visual encoder; a consistency constraint (symmetric KL) couples branch predictions; and an adaptive gated fusion with focal loss yields the final output. On several benchmark datasets, HCP-MER reports competitive/SOTA UF1/UAR across 3/4/7-class settings.

**Strengths:**

1. The proposed method is modular: CLIP text prompts (with learnable tokens), adapter, two visual channels (apex-M and flow), contrastive alignment, JS/KL consistency, and gated fusion, which is easy to follow.

2. It provides clear gains over MER-CLIP on CAS(ME)3 (3/4/7-class).

**Weaknesses:**

1. While the paper presents a seemingly novel Holistic–Componential Prompt (HCP) framework for micro-expression recognition (MER), the methodological contribution over MER-CLIP is rather incremental. The proposed model essentially modifies MER-CLIP by introducing learnable text prompts instead of fixed text descriptions and by separating the prompts into two groups, i.e., holistic (emotion-level) and componential (AU-level). However, this separation does not fundamentally change the underlying vision–language alignment principle. In MER-CLIP, both holistic and AU semantics are already integrated into a single prompt template; thus, HCP-MER can be viewed as a variant that restructures the same information rather than introducing a new learning paradigm.

2. The overall pipeline (CLIP backbone, adapter tuning, optical-flow + magnified apex input, contrastive alignment, and gated fusion) remains largely consistent with existing works, offering limited conceptual novelty. The experimental improvements are moderate and can likely be attributed to the additional learnable prompt parameters or model complexity rather than genuine methodological innovation.

**Questions:**

1. How does the proposed HCP-MER fundamentally differ from MER-CLIP beyond separating the holistic and componential prompts and making them learnable?

2. What new insight or mechanism does this separation introduce that MER-CLIP could not achieve within a unified prompt template?

3.

---

> ### Author Response · Authors · 2025-11-27
> **We thank the reviewer for their comments on the originality and effectiveness of HCP\_MER. We clarified that HCP\_MER differs from MER-CLIP by introducing a hierarchical holistic--componential semantic binding with collaborative inference (consistency loss + adaptive fusion) to address emotion--AU cross-mapping, and ablations show that gains come from this framework rather than added complexity.**
>
> \section*{Rebuttal to Reviewer GpHs}
>
> Dear Reviewer GpHs,
>
> We sincerely thank you for your careful review and insightful comments. Your concerns about the originality of our method have helped us to clarify the core contributions of this work.
>
> \textbf{Reviewer comment:} HCP\_MER appears to be an incremental variant of MER-CLIP, mainly restructuring the same holistic/AU information into learnable and separated prompts without fundamentally changing the underlying vision--language alignment principle.
>
> \textbf{Response:}
> MER-CLIP is indeed a strong baseline that first introduces AU semantics into CLIP. However, it effectively operates with a single-level AU-based prompt: emotion information is only implicitly encoded in a fixed template, and the model infers emotions purely from local muscle movements. As shown by the cross-mapping issue in Fig.~1(a), when different emotions share similar AU combinations (e.g., “anger” vs. “disgust” with AU4), MER-CLIP lacks explicit higher-level semantic guidance, which limits fine-grained MER.
>
> HCP\_MER, in contrast, explicitly builds a hierarchical semantic architecture with two bound prompt groups and two matched visual modalities: (1) holistic prompts provide macro-level emotional context (e.g., “a ME of anger”) and are aligned with motion-magnified frames capturing global expressions; (2) componential prompts encode concrete AU combinations and are aligned with optical-flow maps focusing on local micro-movements. This design lets the model jointly exploit complementary evidence from emotion context and AU-level details, directly targeting the cross-mapping ambiguity rather than simply rephrasing MER-CLIP’s prompt.
>
> Moreover, MER-CLIP performs single-path inference, while HCP\_MER introduces collaborative inference: a symmetric KL-based consistency loss keeps the holistic and componential branches semantically coherent, and an adaptive gated fusion module learns sample-wise weights between them. Thus, the final prediction is the result of dual-path evidence extraction plus semantic calibration and adaptive fusion, rather than a single AU-driven decision pipeline.
>
> \vspace{0.5em}
> \textbf{Reviewer comment:} The pipeline (CLIP backbone, Adapters, optical-flow + magnified apex, contrastive alignment, gated fusion) is largely similar to prior work; gains may mainly come from extra parameters rather than genuine methodological innovation.
>
> \textbf{Response:}
> We agree that it is important to distinguish framework-level innovation from mere parameter increase. While HCP\_MER reuses standard components (CLIP, Adapters, contrastive loss), our main contribution lies in the overall framework: explicit hierarchical semantic binding between holistic and AU-level prompts, modality-specific alignment for each level, and collaborative inference across branches.
>
> Empirically, several results indicate that the gains come from this framework rather than parameter count. Ablation studies show that removing either branch (w/o HOL or w/o COM) leads to a clear performance drop, indicating that the improvement arises from their complementarity, not just from having more parameters. The additional parameters mainly come from lightweight Adapters and learnable prompts, far fewer than full CLIP fine-tuning, yet HCP\_MER outperforms baselines that could use heavier adaptation. Removing the consistency loss or the adaptive fusion also degrades performance, confirming that these modules are necessary for effective dual-branch collaboration rather than redundant complexity.
>
> Finally, the proposed hierarchical semantic binding and collaborative inference are not specific to MER. They provide a general solution pattern for semantic overlap in fine-grained visual recognition and can naturally extend to other tasks that require joint reasoning over macro-level context and micro-level details (e.g., fine-grained categorization, medical imaging). We will make these points more explicit in the revised manuscript. Thank you again for your constructive feedback, which has helped us sharpen and better present the main ideas of our work.

---

### Official Review · Reviewer_BBit · 2025-10-30

**Soundness:** 2
**Presentation:** 1
**Contribution:** 2
**Rating:** 2
**Confidence:** 4

**Summary:**

This paper proposes HCP-MER, which introduces Holistic-Componential Prompt (HCP) Groups to address the issue of semantic ambiguity in emotion recognition. By binding holistic emotional semantics with componential Action Unit (AU) semantics, the method effectively captures richer emotional representations.
Additionally, the model leverages CLIP to establish multi-granularity visual-semantic associations, enhancing cross-modal understanding. A consistency constraint is applied to ensure accurate emotional attribution, while an adaptive gated fusion mechanism combines complementary responses from different branches. Fine-tuned optimization with downstream supervision further improves performance.
Extensive experiments demonstrate the effectiveness of the proposed method, supported by comprehensive ablation studies and visualizations.

**Strengths:**

- The experimental results are strong, with clear gains over baselines.

- This work introduces HCP Groups, which mitigate the emotional semantic ambiguity of single AUs by binding holistic and componential semantic contexts.

**Weaknesses:**

- Writing clarity:
The paper suffers from unclear organization and phrasing. Some crucial details—such as the cross-mapping mechanism, a fundamental part of the method—are relegated to the Appendix (line 054), which hurts readability and comprehension.

- Insufficient analysis:
The issue of cross-contamination (line 058) is mentioned but not deeply analyzed or empirically verified. The claimed mitigation lacks convincing evidence.

- Incremental contributions and scattered focus:
While the components (e.g., multi-granularity complementary associations, consistency constraint, adaptive gated fusion) are technically sound, they appear incremental and loosely connected rather than forming a coherent, central framework. The overall narrative is not compact or consistent, and the contribution feels fragmented.

- Visualization ambiguity:
The paper does not clearly specify what features are used for the Feature Distribution Visualization.
The derivation of attention distribution maps is not explained—how are these obtained, and from which network layer or module?
Figure 5 should include annotations or visual markers to highlight the key findings more explicitly.
It remains unclear how the attention patterns validate the proposed HCP Groups. The connection between these patterns and the HCP mechanism is not sufficiently established, and these attentions might originate from unrelated network components.

**Questions:**

See Weaknesses

---

> ### Author Response · Authors · 2025-11-27
> **We thank Reviewer BBit for their constructive feedback. In our rebuttal, we improve organization by moving the cross-mapping mechanism into the main text, provide stronger evidence that HCP Groups mitigates semantic cross-contamination, clarify our unified framework for resolving semantic ambiguity, and specify the features and attention maps used.**
>
> \section*{Rebuttal to Reviewer BBit}
>
> Dear Reviewer BBit,
>
> We sincerely thank you for your careful reading of our paper and your insightful comments. Your feedback on writing clarity, analysis depth, framework coherence, and visualization is very helpful. Below, we summarize our responses and planned revisions.
>
> \textbf{Reviewer comment:} Writing clarity and organization: the paper is unclearly structured, and key details such as the cross-mapping mechanism are placed in the Appendix.
>
> \textbf{Response:}
> We agree and will improve both structure and wording.
>
> We will move the description and illustration of the “cross-mapping” mechanism from Appendix A into the main text (introduction and/or method section), and present it as a core motivation rather than supplementary material. We will also refine the section flow so that the path from problem definition to method design and experimental validation is more linear and easier to follow. In addition, we will carefully edit the text to remove redundancy, unify terminology, and sharpen technical descriptions.
>
> \textbf{Reviewer comment:} Cross-contamination is mentioned but not deeply analyzed or convincingly supported.
>
> \textbf{Response:}
> We will provide stronger quantitative and category-wise evidence.
>
> On the 7-class CAS(ME)$^{3}$ task, HCP Groups achieves UF1 = 0.5955 and UAR = 0.6047, improving over MER-CLIP (AU-only prompts) by 9.6\% and 10.3\%. Confusion-matrix analysis further shows clear gains on AU-overlapping categories: for “disgust,” true-positive accuracy is 0.83 vs. 0.79 for MER-CLIP; for “anger,” 0.50 vs. 0.41 (21.95\% relative improvement). These results support our claim that HCP Groups, via holistic-componential prompts, mitigate semantic cross-contamination when AU patterns are similar. We will present and discuss these findings explicitly in the revised manuscript.
>
> \textbf{Reviewer comment:} Contributions feel incremental and scattered; components seem loosely connected rather than forming a coherent central framework.
>
> \textbf{Response:}
> Our intention is to provide a unified framework for resolving semantic ambiguity in MER, not a set of independent tricks. We will sharpen this narrative.
>
> In the revision, we will emphasize the following:
> (1) Core problem: complex emotion–AU cross-mapping (many-to-one and one-to-many) leads to semantic ambiguity in MER.
> (2) Core design: HCP Groups binds holistic emotion and componential AU semantics into paired prompts, providing complementary semantic views for each class.
> (3) Supporting mechanisms: multimodal alignment (motion-magnified frames vs. optical flow) supplies the most suitable visual evidence for each semantic level, while consistency regularization and adaptive fusion ensure robust collaboration between the two branches.
>
> We will adjust the introduction, method overview, and conclusion so that this framework-level contribution is consistently highlighted and the paper reads as a single coherent story.
>
> \textbf{Reviewer comment:} Visualization ambiguity: it is unclear what features are used in the feature distribution visualization and how attention maps are obtained and linked to HCP Groups.
>
> \textbf{Response:}
> We will clarify both the construction and interpretation of the visualizations.
>
> For Fig. 4, we will state explicitly that the feature distribution visualization is based on the fused feature $Res_f$, obtained by adaptively gated fusion of $Res_h$ (holistic branch) and $Res_g$ (componential branch).
>
> For Fig. 5, we will specify that attention maps are computed from the attention weights between the [CLS] token and all image patch tokens in the last Transformer block of the CLIP ViT encoder and then upsampled to input resolution. We will then explain the connection to HCP Groups: holistic inputs (motion-magnified frames + holistic prompts) encourage globally distributed facial attention, while componential inputs (optical flow + AU-based prompts) encourage focused attention on local muscle regions. This systematic difference, aligned with the two semantic levels and modalities, provides visual evidence for the intended holistic–componential decomposition. We will enhance captions and add simple visual markers to make these patterns more explicit.
>
> \vspace{0.5em}
> Once again, we thank you for your detailed and constructive feedback. We believe that the revisions described above will substantially improve the paper’s clarity, analytical depth, framework coherence, and the explanatory strength of our visualizations.

---

> > ### Comment · Reviewer_BBit · 2025-11-27
> > **Thank you for your resopnse**
> >
> > I sincerely appreciate the authors’ respectful responses and the revised manuscript. The new version is noticeably clearer than the original. It is recommended to use blue text to highlight the revisions, which would help reviewers more easily identify the changes.
> >
> > Despite the improvements in the revised version, the paper still falls short of the standard required for acceptance. It needs substantial revision in writing quality, presentation, and narrative coherence. One promising direction is that all reviewers recognize the novelty of the Holistic–Componential Prompt (HCP) group design. I encourage the authors to further highlight this idea and provide a deeper, more focused analysis, as they mentioned in their rebuttal by emphasizing the core problem and the core design. In addition, the introduction could benefit from simple quantitative evidence that illustrates how cross-mapping patterns (many to one and one to many) result in semantic ambiguity.
> >
> > Seeing the authors’ respectful responses, the improved version of the paper, and their commitment to further revisions, I am raising my score to weak reject. I hope the authors will continue to strengthen the work by carefully addressing all reviewers’ suggestions.

---

> > > ### Author Response · Authors · 2025-11-27
> > >
> > > We sincerely thank the reviewer for carefully rereading our revised manuscript, for recognizing the improved clarity, and for kindly raising the score. Your encouraging comments and constructive guidance are extremely valuable to us, and we will carefully incorporate them to further improve the quality and impact of this work.

---

### Official Review · Reviewer_9v59 · 2025-11-01

**Soundness:** 2
**Presentation:** 2
**Contribution:** 1
**Rating:** 2
**Confidence:** 4

**Summary:**

This paper presents HCP_MER, a vision-language framework for micro-expression recognition that effectively combines holistic emotional context with fine-grained AU semantics via structured prompt groups. The method demonstrates performance across multiple datasets and granularities, with ablation studies and visual interpretability.

**Strengths:**

1. The holistic-componential prompt groups are well-motivated and effectively address the many-to-many mapping problem between AUs and emotions. The defined
2. The method achieves competitive results across multiple datasets and fine-grained classification tasks, with significant gains in the most challenging 7-class setting.
3. The dual alignment of motion-magnified frames and optical flow with holistic and componential prompts is intuitive and effectively captures both macro and micro facial dynamics.

**Weaknesses:**

1. The final architecture is complex. It requires two distinct visual inputs (Apex\_M
and optical map), processing by two visual encoder branches, two separate prompt
types, two adapters, a consistency loss module, and an adaptive fusion gate. This
introduces many components and hyperparameters compared to a single-branch VLM
approach.
2. While the text prompts are *learnable* (similar to CoOp), the underlying
*structure* of the componential prompt (i.e., which AUs are assigned to the CLASSAU
token for each emotion) appears to be manually defined. This relies on existing facial
anatomy knowledge, and the paper does not specify how these mappings were determined,
especially for emotions with multiple AU variations (the "one-to-many" problem). This
manual engineering step remains a limitation.
3. No cross-dataset evaluation is reported to assess the method's ability to generalize beyond the training distribution. Given the heavy reliance on manually-defined AU mappings, it is unclear whether the approach would transfer to new datasets with different annotation protocols or AU definitions.
4. The choice of adapter depth is dataset-dependent but lacks theoretical or empirical justification beyond data-size heuristics. Furthermore, the paper does not analyze why the consistency constraint and adaptive fusion are necessary if both branches are properly aligned with their respective prompts. The added complexity suggests that the core concept—binding holistic and componential prompts—may not be sufficient on its own.

**Questions:**

1. Have the authors evaluated HCP_MER in a cross-dataset setting? If so, how does it generalize?
2. How was the adapter architecture (e.g., bottleneck dimension, number of layers) chosen? Was it tuned, or based on prior work?
3. Could the method be simplified to use only one visual branch without a significant performance drop? Have the authors studied the relative contribution of each visual modality?

---

> ### Author Response · Authors · 2025-11-27
> **We thank Reviewer 9v59 for their insightful comments. In our rebuttal, we clarify that the dual-branch architecture is a lightweight design on a shared frozen CLIP backbone and ablations show both branches are necessary; that componential prompts are grounded in FACS and dataset AU annotations rather than ad hoc rules; and that new cross-dataset results and ablations on Adapter depth, consistency loss, and adaptive fusion support the generalization and necessity of our design.**
>
> \section*{Rebuttal to Reviewer 9v59}
>
> Dear Reviewer 9v59,
>
> We sincerely thank you for your careful review and insightful comments on model complexity, generalization, and design choices. Your feedback has been very helpful for improving our work. Below we respond to your main concerns.
>
> \textbf{Reviewer comment:} The final architecture is complex, with two visual inputs, two encoder branches, two prompt types, two adapters, a consistency loss, and an adaptive fusion gate.
>
> \textbf{Response:}
> We agree that our architecture is more complex than a single-branch VLM, but this dual-branch design is central to jointly modeling holistic emotion cues and local muscle movements, which are best captured by different visual modalities. Ablation studies (Fig.~3) show that removing either branch (w/o HOL or w/o COM) leads to a clear performance drop, confirming their complementarity. Complexity is controlled by sharing a frozen CLIP ViT backbone across both branches and adding only lightweight Adapters plus small consistency/fusion modules, which we believe is a reasonable trade-off for this fine-grained MER task.
>
> \textbf{Reviewer comment:} The structure of componential prompts (AUs assigned to CLASSAU) seems manually defined and not clearly described, especially for emotions with multiple AU variants.
>
> \textbf{Response:}
> The AU combinations for each emotion are derived from standardized FACS knowledge and prior psychological studies, ensuring a principled basis. Moreover, the ME datasets we use provide AU-region annotations per sample, and our mappings are obtained by aggregating these existing labels rather than ad hoc handcrafting. For emotions with multiple AU realizations, we use dataset-level AU statistics to capture dominant patterns. This process is outlined in Section~3.1 and Appendix~A; in the revision, we will make the mapping procedure and handling of one-to-many emotion–AU relationships more explicit.
>
> \textbf{Reviewer comment:} No cross-dataset evaluation is reported, so generalization beyond the training distribution is unclear, especially given the AU mapping.
>
> \textbf{Response:}
> We fully agree on the importance of cross-dataset evaluation. The initial submission focused on standard within-dataset splits due to differences in acquisition and annotations across ME datasets, which is a limitation. Following your suggestion, we have now conducted cross-dataset experiments (training on CASME II and testing on SAMM, and vice versa). These results and their analysis are included in Table~4 of the revised manuscript and directly evaluate cross-dataset transfer. Combined with our stable LOSO performance and strong results on multiple datasets, these new experiments further support the generalization ability of our method.
>
> \textbf{Reviewer comment:} Adapter depth is chosen per dataset based on size, with limited justification. The necessity of the consistency constraint and adaptive fusion is not fully analyzed if each branch is already aligned with its prompts.
>
> \textbf{Response:}
> Adapter depth is chosen using a standard capacity–overfitting trade-off in transfer learning: shallow Adapters (1 layer) for smaller datasets to reduce overfitting, and deeper Adapters (3 layers) for CAS(ME)$^{3}$ to better exploit its larger scale. We will clarify this rationale in the revision.
>
> Even if each branch is aligned with its own prompts, they provide different perspectives (global vs. local). The consistency constraint acts as a semantic calibrator, encouraging coherent predictions and avoiding contradictions between branches, while the adaptive fusion gate allows sample-wise adjustment of the relative importance of global and local cues. Ablations (e.g., removing CC or fusion) lead to noticeable performance drops, showing that these modules are necessary to make the dual-branch design work effectively rather than being redundant complexity. Ablations also show that collapsing to a single branch significantly harms performance, supporting the necessity of the dual-modality design.
>
> \vspace{0.5em}
> Once again, we sincerely thank you for your rigorous and constructive comments. They have helped us refine and strengthen this work, and we believe that the clarifications and new experiments in the revised version will further improve the rigor and contribution of the paper.

---

### Official Review · Reviewer_7ra7 · 2025-11-01

**Soundness:** 2
**Presentation:** 2
**Contribution:** 1
**Rating:** 4
**Confidence:** 3

**Summary:**

Micro-expressions (ME) are characterized by their brief duration, low intensity, and localized nature, with subtle differences between emotion classes. Low-level visual features and single-type text descriptions (e.g., "a photo of [class]") struggle to capture higher-order emotional meanings, while the many-to-many relationships between Action Units (AUs) and emotions lead to semantic confusion. This paper introduces the "HCP (Holistic-Componential Prompt) group" based on CLIP-style VLMs, pairing holistic emotional meanings with componential movement descriptions. For each emotion category, it establishes one-to-one connections between holistic emotion descriptions and corresponding AU descriptions, reducing semantic ambiguity.

**Strengths:**

- The design of the HCP group is novel. By pairing holistic emotional meanings with componential AU meanings in a "HCP group," it effectively reduces semantic confusion that occurs with single prompts.
- The method demonstrates high effectiveness on CASME II/SAMM/SMIC/CAS(ME)3 datasets, particularly showing a significant improvement of approximately 10 percentage points compared to MER-CLIP on the 7-class CAS(ME)3 dataset.

**Weaknesses:**

- The technical novelty is limited. While acknowledging the novelty in the HCP group design, the proposed method achieves high performance through highly specialized engineering focused on micro-expression recognition. Although this has its merit, ICLR as a top-tier machine learning and deep learning conference requires technical innovations that can be transferred to other machine learning applications. This paper, while excellent as an application use case, would be more suitable for domain-specific conferences like FG.
- The writing needs improvement. There are numerous instances where paragraph writing fails to provide logical explanations effectively (e.g., Section 3.4). Additionally, the paper appears unpolished due to repetitive content, detailed information that should be in the appendix (such as GPU specifications), excessive white space, and unclear figures (e.g., Fig. 4).
- There are concerns about reproducibility. The number of Adapter layers is only described as variable according to dataset size (single layer for small datasets, multiple layers for CAS(ME)3). The documentation is insufficient regarding specific settings for $\tau$, prompt length $k$, Adapter bottleneck dimensions, Focal loss parameters, MagNet and FlowNet configurations, and input resolution and preprocessing procedures.
- (Minor comment) Non-variable subscripts should be typeset in roman font.

**Questions:**

- How were hyperparameters such as learning rate, batch size, and $\lambda_1$ and $\lambda_2$ determined?

---

> ### Author Response · Authors · 2025-11-27
> **We thank Reviewer 7ra7 for their thoughtful comments on generality, writing, and reproducibility. In our rebuttal, we clarify that our method is a general multi-granularity vision–language framework rather than a MER-only engineering tweak, and we commit to substantially improving the paper’s structure and exposition. We also ensure full reproducibility by clearly documenting implementation choices and fixing all remaining typographical issues.**
>
> \section*{Rebuttal to Reviewer 7ra7}
>
> Dear Reviewer 7ra7,
>
> We sincerely thank you for your valuable comments and suggestions. Your feedback helped us identify key shortcomings and provided clear directions for improvement. Below we respond to your main concerns and outline our planned revisions.
>
> \textbf{Reviewer comment:} The method, though novel, is specialized for MER and seems like an engineering solution with limited transferability.
>
> \textbf{Response:}
> Our goal is to propose a generally useful multi-granularity vision–language framework, not a MER-only engineering trick.
>
> (1) By binding holistic emotion semantics with local AU semantics, HCP Groups addresses semantic ambiguity from complex emotion–AU mappings. The same holistic–component prompt idea can be extended to other fine-grained tasks (e.g., macro-expressions, visual affect analysis, or generic category–attribute recognition with LLM-generated attributes).
> (2) Aligning motion-magnified frames with holistic prompts and optical flow with componential prompts defines a general spatio-temporal alignment pattern that can transfer to other dynamic tasks (e.g., person re-ID with appearance vs. gait, dynamic gesture recognition with global pose vs. local joints).
> (3) The symmetric KL consistency loss and adaptive gating are generic mechanisms for coordinating multi-branch predictions and can be reused in multi-branch, multi-task, or ensemble models.
>
> We will make this broader applicability and connection to CLIP/CoOp-style VLMs more explicit in the revision.
>
> \textbf{Reviewer comment:} Writing quality and presentation need improvement (unclear logic, repetition, low-level details in main text, excessive white space, unclear figures).
>
> \textbf{Response:}
> We agree and will:
>
> – Rewrite Section 3.4 and related parts to more clearly motivate the consistency loss and clarify the logic between holistic/local branches.
> – Remove redundant statements and move low-level details (e.g., GPU specs) to Appendix C so the main text focuses on key methods and results.
> – Redraw figures, especially Fig.~4, with higher resolution and clearer annotations.
> – Strictly follow the ICLR template to reduce unnecessary white space and improve layout.
>
> \textbf{Reviewer comment:} Adapter depth is only described by dataset size, and key implementation details ($\tau$, $k$, bottleneck, focal loss, MagNet/FlowNet configs, resolution, preprocessing) are missing.
>
> \textbf{Response:}
> We thank you for pointing out this reproducibility gap. In the revised version, we provide full hyperparameter and configuration details (now in Appendix C) and briefly summarize them in the method section, including:
>
> – Contrastive temperature $\tau = 0.01$; prompt length $k = 8$ for all datasets; Adapter bottleneck 64 for SMIC/CASME II/SAMM and 128 for CAS(ME)$^{3}$; focal loss with $\gamma = 2.0$, $\alpha = 0.25$.
> – MagNet with magnification factor 2 and pre-trained FlowNet2.0 without fine-tuning.
> – All visual inputs resized to $224\times224$ and normalized with ImageNet mean/std.
>
> We will present these settings in a compact table to support reproducibility.
>
> \textbf{Reviewer comment:} How are learning rate, batch size, and $\lambda_1$, $\lambda_2$ chosen?
>
> \textbf{Response:}
> All key hyperparameters are tuned via systematic experiments: we use a learning rate of $1\times10^{-4}$, batch size 32, and cosine annealing for stable convergence. For loss weights, we set $\lambda_1 = 0.5$, $\lambda_2 = 0.5$ and perform sensitivity analysis over $\{0.01, 0.05, 0.1, 0.5, 1\}$; we will include these results in the appendix.
>
> \textbf{Reviewer comment:} Non-variable subscripts should be in roman (upright) font.
>
> \textbf{Response:}
> We appreciate this typographical correction and will carefully review all formulas to ensure non-variable subscripts, function names, and similar symbols use roman font, following standard conventions.
>
> In summary, we will (1) clarify the broader applicability of our framework, (2) improve writing and organization, (3) provide complete implementation details, and (4) correct typesetting issues. We respectfully ask you to reconsider your evaluation in light of these revisions, and we thank you again for your thoughtful and constructive feedback.

---

> > ### Comment · Reviewer_7ra7 · 2025-11-27
> > **Thank you for your resopnse**
> >
> > I sincerely appreciate the authors' respectful responses. While I understand the logic behind your claims regarding transferability, I consider that such claims can only be substantiated through actual application to different tasks or, at the very least, by specifying concrete tasks and methodologies.
> >
> > Regarding the writing quality, although some improvements have been made, the paragraphs remain poorly organized, particularly in the introduction, resulting in sentences that fail to effectively convey the motivation behind this work.
> >
> > Considering the above reasons, I maintain my original scoring. I hope my comments will help with your future submissions.

---

> > > ### Author Response · Authors · 2025-11-27
> > >
> > > We sincerely thank Reviewer 7ra7 for the follow-up response and for the time and care invested in evaluating our work. We fully understand and respect your decision to maintain the original score, and we appreciate your clear suggestions regarding demonstrating transferability through concrete additional tasks and further improving the structure and motivation in the introduction. Your feedback has been very helpful for us to reflect on both the scope of our experiments and our writing, and we will carefully incorporate these insights into future revisions and submissions. Thank you again for your constructive and thoughtful comments.

---

### Meta-Review · Area_Chair_cZsH · 2025-12-04

**Summary:**

The submission received all negative ratings. It proposes a vision-language framework for micro-expression recognition that uses holistic-componential prompt groups to address semantic ambiguity between emotions and facial action units. While reviewers acknowledged the novelty of the prompt group design and competitive experimental results, they raised significant concerns about the paper’s technical novelty, generalizability, presentation clarity, and overall contribution to the machine learning community. The key criticisms included the limited methodological novelty beyond engineering adjustments, insufficient evidence of transferability to other tasks, unclear writing and organization, lack of cross-dataset validation, and inadequate analysis of core components such as the consistency constraint and adaptive fusion mechanism.

**Reviewer Concerns:**

The authors provided a rebuttal to illustrate additional implementation details (hyperparameters, adapter configurations) to improve reproducibility; they also carried out cross-dataset experiments and included them in the revised manuscript. They promise to improve writing quality and visualization clarity.

There are still outstanding issues. The method remains highly specialized for MER and does not demonstrate broader applicability or fundamental technical advancement suitable for ICLR (shared by 7ra7, GpHs). Reviewer BBit still doubted the technical depth and novelty. The necessity of dual branches, consistency loss, and adaptive fusion was not convincingly justified.

**Reviewer Scores:**

Reviewer 7ra7 insisted on the original rating (4) after discussion with the authors. Reviewer BBit promised to raise the score from 2 to 4 after the discussion. Reviewers 9v59 and GpHs rated as 2. They did not say anything about the change of rating. While I believe it is hard for them to raise scores, considering all the negative ratings.

---

### Decision · Program_Chairs · 2026-01-26

Reject